# Characteristics of the right ventricle in left ventricular noncompaction with reduced ejection fraction in the light of dilated cardiomyopathy

**Zsófia Gregor, Anna Réka Kiss, Kinga Grebur(iD), Zsófia Dohy(iD), Attila Kovács(iD), Béla Merkely(iD), Hajnalka Vágó, Andrea Szűcs(iD)** *

Heart and Vascular Center of Semmelweis University, Budapest, Hungary

* szucsand@gmail.com

## Abstract

### Background

Reports of left ventricular noncompaction (LVNC) rarely include descriptions of the right ventricle (RV). This study aimed to describe the characteristics of the RV in LVNC patients with reduced LV function (LVNC-R) compared with patients with dilated cardiomyopathy (DCM) and subjects with LVNC with normal left ventricular ejection fraction (LV-EF) (LVNC-N).

### Methods

Forty-four LVNC-R patients, 44 LVNC-N participants, and 31 DCM patients were included in this retrospective study (LV-EF: LVNC-R: 33.4±10.2%; LVNC-N: 65.0±5.9%; DCM: 34.6 ±7.9%). Each group was divided into two subgroups by the amount of RV trabeculation.

### Results

There was no difference in the RV-EF between the groups, and the RV trabecular mass correlated positively with the RV volume and negatively with the RV-EF in all the groups. All the measured parameters were comparable between the groups with decreased LV function. The hypertrabeculated RV subgroups showed significantly higher RV volumes and lower RV-EF only in the decreased-LV-function groups. The correlation of LV and RV trabeculation was observed only in the LVNC-N group, while LV trabeculation correlated with RV volumes in both noncompacted groups. Both decreased-LV-function groups had worse RV strain values than the LVNC-N group; however, RV strain values correlated with RV trabeculation predominantly in the LVNC-R group.

### Conclusions

The presence and characteristics of RV hypertrabeculation and the correlations between LV trabeculation and RV parameters raise the possibility of RV involvement in noncompaction;

**Data Availability Statement:** All relevant data are within the paper and its Supporting Information files.

**Funding:** Thematic Excellence Program (Tématerületi Kiválósági Program, 2020-4.1.1.-TKP2020) of the Ministry for Innovation and Technology in Hungary within the framework of the Therapeutic Development and Bioimaging Programs of Semmelweis University (ZG, ARK, ZD). https://2015-2019.kormany.hu/hu/innovacios-es-technologiai-miniszterium National Research, Development and Innovation Office, the Development of Scientific Workshops of Medical, Health Sciences and Pharmaceutical Education (Project identification number: EFOP-3.6.3-VEKOP-16-2017-00009 (ZG, ARK, ZD). https://nkfih.gov.hu/about-the-office Project no. NVKP_16-1–2016-0017 (National Heart Program) has been implemented with the support provided by the National Research, Development, and Innovation Fund of Hungary, financed under the NVKP_16 funding scheme (ZG, ARK, ZD). https://nkfih.gov.hu/about-the-office The funders had no role in study design, data collection and analysis, decision to publish, or preparation of the manuscript.

**Competing interests:** The authors have declared that no competing interests exist.

moreover, RV strain values might be helpful in the early detection of RV function deterioration.

## Introduction

Many investigations have highlighted the characteristics of the right ventricle (RV) in physiological and pathological conditions, but the connection between the RV and the left ventricle (LV) must be emphasized, as they are inseparable through their direct mechanical interactions [1, 2].

Although some studies have revealed the relationship between RV and LV trabeculation and the effect of LV trabeculation on RV parameters in various conditions, the RV characteristics in the case of a hypertrabeculated left ventricle, e.g., noncompaction (LVNC), are less well described. In an LVNC population with good LV ejection fraction (EF), the characteristics of RV trabeculation suggested the involvement of the RV in noncompaction [3, 4]. However, it is still unknown how the deterioration of LV-EF determines RV characteristics in noncompaction patients. As the involvement of the RV in noncompaction predicts a worse prognosis even in patients with good LV function, presumably, this might be predictable in cases of reduced LV function, and these patients require even more accurate follow-up [4].

Our study aimed to describe the characteristics of the RV and quantify the connection among RV parameters, LV function, and LV trabeculation in LVNC patients with reduced LV function (LVNC-R); furthermore, we compared these correspondences with those in both dilated cardiomyopathy (DCM) patients and LVNC subjects with normal EF (LVNC-N).

## Methods

### Study population

Forty-four LVNC-R patients (EF<50%), 44 LVNC-N participants (EF>50%), and 31 DCM patients were included in this retrospective study.

Both LVNC-R and LVNC-N were diagnosed based on the morphological criteria of Petersen (noncompact/compact layer ratio >2.3 in end-diastolic, long-axis views). For better patient selection, Jacquier's criterion (trabeculated LV mass is >20% of the total LV myocardial mass) was 42% in the LVNC-R group and 36% in the LVNC-N group [5, 6]. In addition to these morphological criteria, at least one clinical symptom was described in the anamnestic details in all of the LVNC participants as recommended by Vergani et al. [7] (S1 Table).

The enrolled DCM patients met the following diagnostic criteria: dilated LV chamber, increased LV volumes, and decreased systolic LV function (EF<50%) with the exclusion of other causes of LV dysfunction [8].

The following exclusion criteria were applied in all three groups: patients with poor image quality or whose scans were performed after contrast agent administration or contained arrhythmic or respiratory artifacts were excluded, as it would have modified the postprocessing evaluation [9]. The presence of any ischemic, valvular, or congenital heart diseases or other coexisting cardiomyopathies (non-DCM and non-LVNC); other relevant comorbidities (hypertension, diabetes mellitus, endocrine disorders, chronic kidney or systemic diseases, etc.) or intense sports activity (>6 hours/week) were criteria for exclusion from the study (S2 Table). The baseline characteristics of the studied groups are listed in Table 1.

To study the hypertrabeculation of the RV, all the groups were divided into two subgroups by the amount of RV trabeculation. The patients were evaluated individually based on their

**Table 1. Baseline characteristics and left ventricular parameters of the studied populations.**

| | DCM | LVNC-R | LVNC-N | p (LVNC-R vs. DCM) | p (LVNC-R vs. LVNC-N) | p (DCM vs. LVNC-N) |
|---|---|---|---|---|---|---|
| **Number of patients** (males) | 31 (18) | 44 (30) | 44 (30) | 0.369 | 1 | 0.369 |
| **Age** (years) | 51.3±14.8 | 55.4±11.0 | 45.8±13.3 | 0.360 | **0.002** | 0.173 |
| **BSA** ($m^2$) | 1.99±0.23 | 1.96±0.26 | 1.98±0.22 | 0.861 | 0.931 | 0.979 |
| **BMI** ($kg/m^2$) | 27.7±5.1 | 26.5±5.5 | 26.3±3.7 | 0.517 | 0.991 | 0.445 |
| **LGE** n (%) | 28 (71.4) | 36 (63.9) | 1 (2.7) | 0.291 | **0.0001** | **0.0001** |
| **LV-EDVi** ($ml/m^2$) | 112.9±32.3 | 114.2±30.5 | 74.9±13.8 | 0.772 | **0.0001** | **0.0001** |
| **LV-ESVi** ($ml/m^2$) | 75.5±28.5 | 78.1±29.9 | 26.6±7.8 | 0.782 | **0.0001** | **0.0001** |
| **LV-SVi** ($ml/m^2$) | 37.4±8.2 | 36.6±10.1 | 48.5±8.0 | 0.917 | **0.0001** | **0.0001** |
| **LV-EF** (%) | 34.6±7.9 | 33.4±10.2 | 65.0±5.9 | 0.806 | **0.0001** | **0.0001** |
| **LV-TMi** ($g/m^2$) | 43.6±9.7 | 51.3±13.4 | 28.0±8.0 | **0.008** | **0.0001** | **0.0001** |
| **LV-CMi** ($g/m^2$) | 71.1±20.7 | 69.2±15.5 | 50.4±11.8 | 0.866 | **0.0001** | **0.0001** |
| **LV-GLS** (%) | -10.9±3.8 | -12.1±3.7 | -21.3±2.2 | 0.286 | **0.0001** | **0.0001** |
| **LV-GCS** (%) | -15.3±6.3 | -14.0±4.8 | -30.2±5.1 | 0.587 | **0.0001** | **0.0001** |

DCM: dilated cardiomyopathy, LVNC-R: left ventricular noncompaction with reduced LV function, LVNC-N: left ventricular noncompaction with good LV function, BSA: body surface area, BMI: body mass index, LGE: late gadolinium enhancement, LV-EDVi: left ventricular end-diastolic volume index, LV-ESVi: left ventricular end-systolic volume index, LV-SVi: left ventricular stroke volume index, LV-EF: left ventricular ejection fraction, LV-TMi: left ventricular end-diastolic trabecular and papillary muscle mass index, LV-CMi: left ventricular end-diastolic compact myocardial mass index, LV-GLS: left ventricular global longitudinal strain, LV-GCS: left ventricular global circumferential strain

The bold values indicate statistical significance (p<0.05).

sex- and age-specific reference values, and those who exceeded that range were categorized as patients with RV hypertrabeculation (HT), while those who were within the abovementioned reference range were classified as having normal RV trabeculation (NT) [10].

All procedures in this investigation followed the 1964 Declaration of Helsinki and its later amendments or comparable ethical standards. Ethical approval was obtained from the Central Ethics Committee of Hungary. Participants provided written informed consent, as they had to sign a form before the examination that their results could be used for scientific purposes.

## Image acquisition and analysis

Cardiac MRI examinations were performed on a 1.5 T MRI scanners (Magnetom Aera, Siemens Healthineers, Erlangen, Germany; or Achieva, Philips Medical System, Eindhoven, the Netherlands). Retrospectively gated, balanced steady-state free precession (bSSFP) cine images were acquired in conventional two-chamber, three-chamber and four-chamber long-axis views. Breath-hold short-axis cine images from base to apex were obtained with full coverage of the LV and RV.

The following scanning parameters were similar for both Siemens and Philips scanners: repetition time: 2.5 and 2.7 ms, respectively; echo time: 1.15 and 1.3 ms, respectively; flip angle: 58˚ and 60˚, respectively; spatial resolution: 1.5 ×1.5 mm for both scanners; and temporal resolution: 25 frames per cardiac cycle for both scanners. The slice thickness was 8 mm with no interslice gap, and the field of view was 350 mm on average adapted to body size. All of the images were obtained before the administration of the contrast agent (gadobutrol, 0.15 ml/kg, if given).

For the postprocessing analysis, Medis Suite software version 3.2 (Medis Medical Imaging Systems) and its modules QMass and QStrain were used. Semiautomatic tracing with manual correction of the endo- and epicardial contours of the left and right ventricles was performed

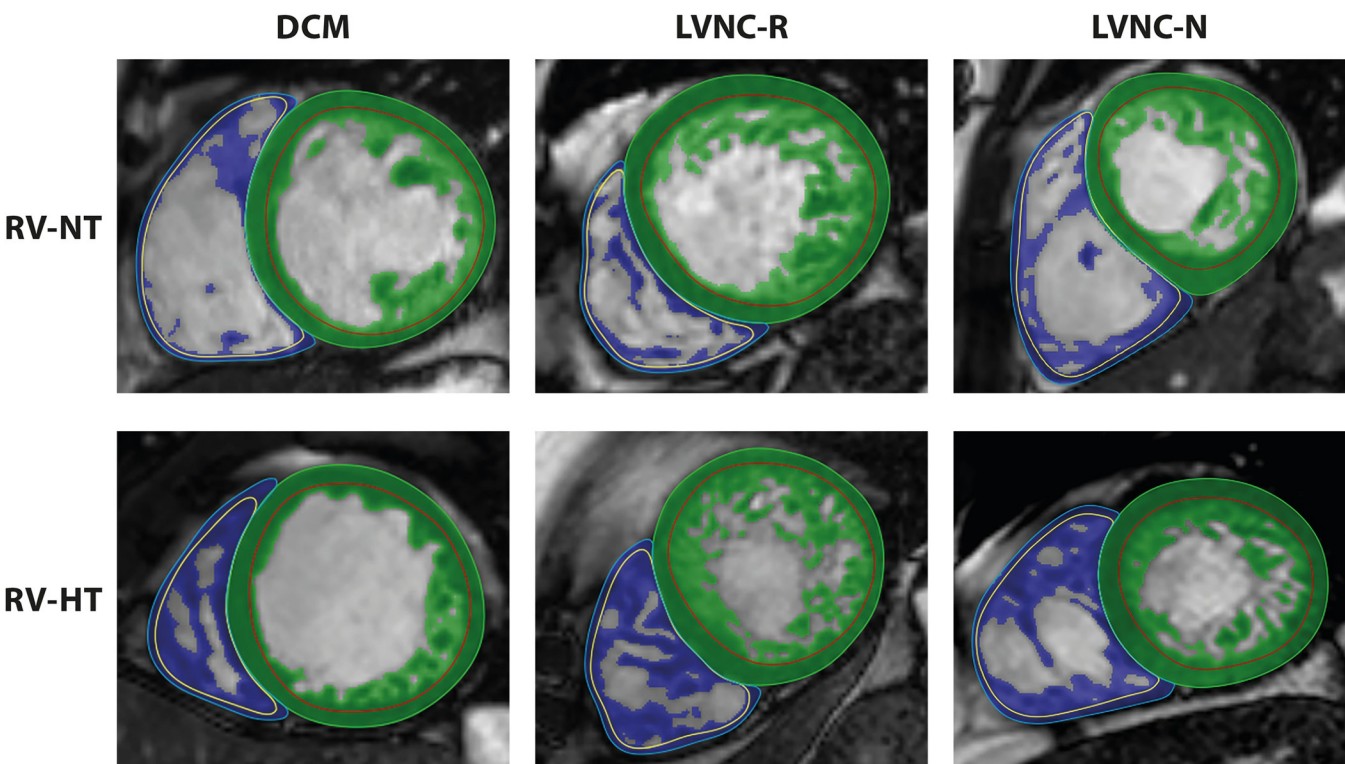

**Fig 1. Representative cardiac MRI images of the studied populations in short axis view with the applied contouring technique and trabecular quantification.** Conventional, manually corrected semiautomatic contours with the threshold-based trabecular quantification method: the green area represents the left ventricular trabeculation, and the blue area represents the right ventricular trabeculation. DCM: dilated cardiomyopathy, LVNC-N: left ventricular noncompaction with good LV function, LVNC-R: left ventricular noncompaction with reduced LV function, RV-HT: right ventricle with hypertrabeculation, RV-NT: right ventricle with normal trabeculation. red line: left ventricular endocardial border, green line: left ventricular epicardial border, yellow line: right ventricular endocardial border, blue line: right ventricular epicardial border.

at end-systole and end-diastole. These contours were made by two independent observers (ZG with 6 years of experience and AS with 11 years of experience).

The LV and RV volumetric, functional and myocardial mass parameters were calculated with the MassK algorithm of the QMass module. Based on the altered signal intensities, this threshold-based papillary and trabeculated muscle quantification algorithm automatically identifies all voxels within the epicardial contour as blood or myocardium; therefore, the detected myocardial voxels within the endocardial borders include the papillary and trabeculated muscles (Fig 1). For the calculation of the measured parameters, the threshold was set to the default and was not changed during the postprocessing analysis. Images performed after contrast agent administration or containing artifacts were not considered due to their poor analyzability.

Based on the above, the LV and RV end-diastolic volume (LV-EDV, RV-EDV), end-systolic volume (LV-ESV, RV-ESV), stroke volume (LV-SV, RV-SV), ejection fraction (LV-EF, RV-EF), end-diastolic total myocardial mass and end-diastolic trabecular and papillary muscle mass (LV-TM, RV-TM) were calculated. The compact myocardial mass (LV-CM, RV-CM) was calculated as the difference between the total myocardial mass and the LV-TM and RV-TM. All of the measured parameters were indexed to body surface area (i).

For the LV and RV deformation analysis, the feature-tracking technique was used within the QStrain module of the software. Endocardial strain values were calculated based on the manually traced endocardial contours in end-diastole and end-systole in the short axis and the

2-, 3- and 4-chamber long-axis images of the left ventricle and in the 4-chamber view of the right ventricle. The LV global longitudinal strain (LV-GLS), LV global circumferential strain (LV-GCS), RV global longitudinal strain (RV-GLS), RV free-wall strain (RV-FWS) and RV septal strain (RV-SS) were measured.

## Statistical analysis

Continuous variables are represented as the mean and standard deviation (SD), and discrete parameters are described as counts and percentages. The Shapiro–Wilk test was used to assess the normality of distributions. For the comparison of the three groups, one-way analysis of variance (ANOVA) was used with Tukey's post-hoc test in the case of normal distribution and equal variances. The Kruskal–Wallis test was used to compare nonnormally distributed data. An independent-samples t-test or the Mann–Whitney test was performed to compare the sub-groups, as appropriate. Linear correlations between the studied parameters were assessed with Pearson's or Spearman's correlation coefficient. To compare categorical variables, the chi-squared test was performed. The intraclass correlation coefficient (ICC) and 95% confidence intervals were used to estimate the interobserver agreement. A p value <0.05 was considered statistically significant. IBM SPSS Statistics (Version 25.0, Armonk, NY) was used for calculations. Figures were generated with GraphPad Prism version 8.0.0 on Windows (GraphPad Software, San Diego, California USA).

## Results

The ICCs were calculated to describe the interobserver agreement on the LV and RV parameters, and excellent values were measured for all parameters (S3 Table).

Contrast agent was used for 84.0% of the studied patients (n = 100), and late gadolinium enhancement (LGE) with a nonischemic pattern was present in 71.4% of the DCM, 63.9% of the LVNC-R and 2.7% of the LVNC-N groups (Table 1).

First, we presented the baseline characteristics of the LV. The LVNC-R and LVNC-N groups were different in all of the measured parameters, while the LVNC-R and DCM groups differed significantly only in the LV-TMi. LV-TMi correlated negatively with LV-EF (LVNC-R: r = -0.680, p = 0.0001; DCM: r = -0.431, p = 0.016; LVNC-N: r = -0.434, p = 0.003). The LVNC-N group had significantly better LV-GLS and LV-GCS values than the reduced-EF group (Table 1).

The measured RV volumetric parameters were all comparable between the LVNC-R and DCM groups, while the RV-EDVi and RV-SVi values were significantly higher in the LVNC-N group than in the LVNC-R or DCM groups. It is remarkable that independent of the LV-EF, the RV-EF was in the normal range in all of the studied groups, and no significant differences were observed between the three groups. Furthermore, there were no significant differences in either the RV-TMi or RV-CMi between the observed groups (Table 2).

Next, the patients were divided by RV trabecular mass into two subgroups: RV-HT and RV-NT (Table 2). RV-HT was present in equal proportions in the LVNC-N and DCM groups, while more patients showed a higher-than-normal amount of RV trabeculation in the LVNC-R group than in the DCM group. However, more patients presented with reduced RV-EF in the DCM group than in the noncompacted groups [11].

In the comparison of the HT and NT subpopulations, not only the RV-TMi but also the RV-CMi were significantly higher in the HT groups. We did not find differences in the RV volumetric or functional parameters between the NT and HT subgroups of the LVNC-N population. In contrast, RV volumetric parameters were significantly higher and RV-EF was significantly lower in the decreased-LV-function groups with RV-HT. Unlike the other two

**Table 2. Right ventricular parameters of the studied populations and the prevalence of the hypertrabeculated subgroups in each population.**

| | DCM | LVNC-R | LVNC-N | p (LVNC-R vs. DCM) | p (LVNC-R vs. LVNC-N) | P (DCM vs. LVNC-N) |
|---|---|---|---|---|---|---|
| **RV-EDVi** (ml/m$^2$) | 61.0±17.2 | 61.0±16.4 | 70.8±15.2 | 1.00 | **0.015** | **0.03** |
| **RV-ESVi** (ml/m$^2$) | 25.4±11.5 | 25.5±11.8 | 27.3±7.3 | 0.998 | 0.775 | 0.775 |
| **RV-SVi** (ml/m$^2$) | 34.0±9.2 | 34.7±9.5 | 43.5±8.0 | 0.936 | **<0.0001** | **<0.0001** |
| **RV-EF** (%) | 57.8±14.0 | 58.7±12.6 | 61.7±7.5 | 0.944 | 0.433 | 0.320 |
| **RV-TMi** (g/m$^2$) | 22.4±6.4 | 23.9±8.9 | 22.4±6.8 | 0.659 | 0.634 | 0.999 |
| **RV-CMi** (g/m$^2$) | 14.1±4.4 | 14.6±3.8 | 14.7±4.2 | 0.842 | 0.997 | 0.808 |
| **RV-GLS** (%) | -19.8±5.7 | -20.0±7.1 | -24.2±4.1 | 0.99 | **0.003** | **0.005** |
| **RV-FWS** (%) | -21.6±9.1 | -23.0±9.1 | -28.3±5.0 | 0.721 | **0.006** | **0.005** |
| **RV-SS** (%) | -13.0±7.3 | -11.3±6.4 | -14.1±4.8 | 0.477 | **0.024** | 0.710 |
| **Patients with RV hypertrabeculation (HT)** n (%) | 15 (48.4) | 23 (52.3) | 21 (47.7) | 0.740 | 0.670 | 0.955 |
| **Reduced RV function in the HT subgroup n** (%) | 6 (40.0) | 8 (34.8) | 2 (9.5) | 0.744 | 0.117 | 0.079 |

DCM: dilated cardiomyopathy, LVNC-R: left ventricular noncompaction with reduced LV function, LVNC-N: left ventricular noncompaction with good LV function, RV-EDVi: right ventricular end-diastolic volume index, RV-ESVi: right ventricular end-systolic volume index, RV-SVi: right ventricular stroke volume index, RV-EF: right ventricular ejection fraction, RV-TMi: right ventricular end-diastolic trabecular and papillary muscle mass index, RV-CMi: right ventricular end-diastolic compact myocardial mass index, RV-GLS: right ventricular global longitudinal strain, RV-FWS: right ventricular free-wall strain, RV-SS: right ventricular septal strain, RV: right ventricle, HT: right ventricle hypertrabeculation

The bold values indicate statistical significance (p<0.05).

groups, the HT subgroup of the LVNC-R group showed significantly worse RV strain values (RV-GLS, RV-FWS, RV-SS) than the NT subgroup (Fig 2 and S4 Table).

Regarding the deformation analysis of the RV in the total population, there were no significant differences in the RV-GLS and RV-FWS between the reduced-LV-function groups; however, relevant differences were found between the good- and decreased-LV-EF populations. RV-SS differed significantly only between the two LVNC groups (Table 2).

The RV-TMi showed a strong positive correlation with the RV volumes and a negative correlation with the RV-EF in all of the studied groups. Further relevant correlation between the RV-TMi and LV-TMi were observed only in the LVNC-N group, while this observation was not present in the decreased-LV-function groups. Notably, the LV-TMi showed more significant correlations with RV parameters in the LVNC groups than in the DCM group. The RV-EF was correlated with all RV parameters in all the groups, while the LV-EF showed a correlation with volumes only in the LVNC-R group (Table 3).

All the measured RV strain values showed significant correlations with the RV-TMi and LV-TMi and with the RV-EF, especially in the LVNC-R group. The RV-GLS and RV-FWS showed negative correlations with the LV-EF in the reduced LV function groups, and the RV-SS correlated with the LV-TMi, RV-TMi, RV-EF, RV-EDVi and RV-ESVi only in the LVNC-R group (Table 3).

## Discussion

In this study, we described the RV trabeculation and RV functional characteristics of patients with LVNC and decreased LV function.

### LV parameters

After studying the characteristics of the LV, the good- and decreased-LV-function groups differed in the volumetric and functional parameters, while the only difference between the LVNC-R and DCM populations was in the value of the LV-TMi. These findings were

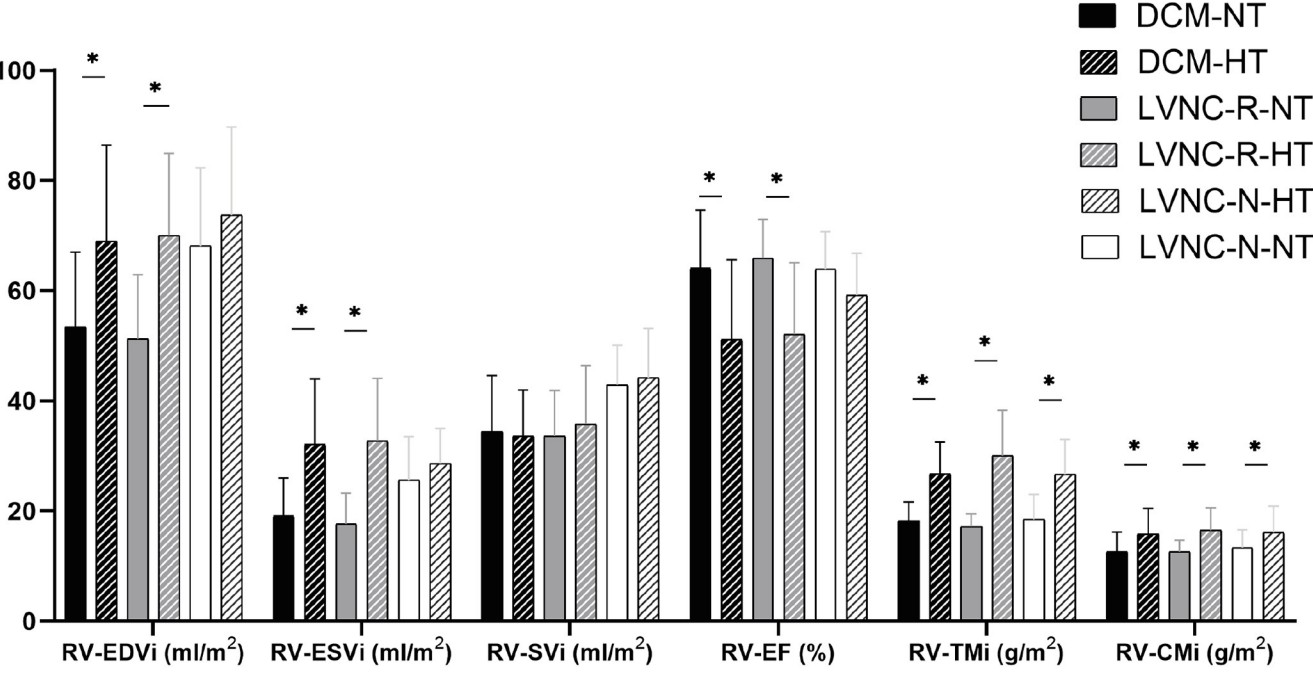

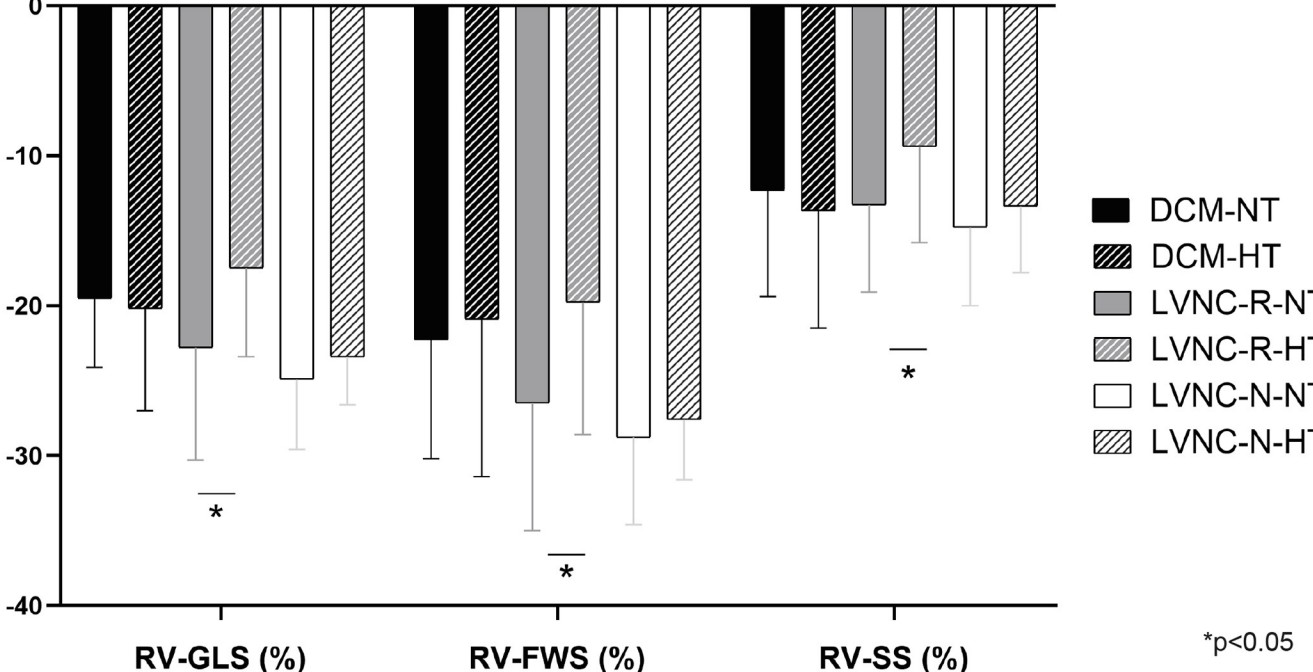

**Fig 2. Comparison of the subgroups with normal RV trabeculation (NT) and with RV hypertrabeculation (HT) within the groups (values of the represented parameters are shown in S4 Table).** DCM-HT: dilated cardiomyopathy with right ventricular hypertrabeculation, DCM-NT: dilated cardiomyopathy with normal right ventricular trabeculation, LVNC-N-HT: left ventricular noncompaction with good left ventricular function and right ventricular hypertrabeculation, LVNC-N-NT: left ventricular noncompaction with good left ventricular function and normal right ventricular trabeculation, LVNC-R-HT: left ventricular noncompaction with reduced left ventricular function and right ventricular hypertrabeculation, LVNC-R-NT: left ventricular noncompaction with reduced left ventricular function and normal right ventricular trabeculation, RV-CMi: right ventricular end-diastolic compact myocardial

mass index, RV-EDVi: right ventricular end-diastolic volume index, RV-EF: right ventricular ejection fraction, RV-ESVi: right ventricular end-systolic volume index, RV-FWS: right ventricular free-wall strain, RV-GLS: right ventricular global longitudinal strain, RV-SS: right ventricular septal strain, RV-SVi: right ventricular stroke volume index, RV-TMi: right ventricular end-diastolic trabecular and papillary muscle mass index.

consistent with the literature in all aspects [12–16]. For a comprehensive understanding of LVNC, we should also mention that according to our previous studies, both LV and RV volumetric, functional and myocardial mass parameters of a large cohort LVNC population with good LV-EF also differed significantly from healthy controls [4, 17].

**Table 3. Correlations of the right ventricular parameters.**

**A) Correlations between RV parameters and LV and RV trabeculation**

|  | LV-TMi | | | | | | RV-TMi | | | | | |
|---|---|---|---|---|---|---|---|---|---|---|---|---|
|  | DCM | | LVNC-R | | LVNC-N | | DCM | | LVNC-R | | LVNC-N | |
|  | r | p | r | p | r | p | r | p | r | p | r | p |
| **RV-EDVi** | 0.281 | 0.126 | **0.370** | **0.014** | **0.502** | **0.001** | **0.637** | **0.0001** | **0.714** | **0.0001** | **0.539** | **0.0001** |
| **RV-ESVi** | 0.061 | 0.746 | **0.362** | **0.016** | **0.443** | **0.003** | **0.814** | **0.0001** | **0.816** | **0.0001** | **0.646** | **0.0001** |
| **RV-SVi** | **0.411** | **0.022** | 0.023 | 0.881 | **0.402** | **0.007** | 0.073 | 0.695 | 0.109 | 0.48 | 0.316 | 0.037 |
| **RV-EF** | 0.129 | 0.49 | -0.253 | 0.097 | -0.202 | 0.188 | **-0.626** | **0.0001** | **-0.695** | **0.0001** | **-0.542** | **0.0001** |
| **RV-TMi** | 0.231 | 0.212 | 0.285 | 0.061 | **0.469** | **0.001** | 1 | - | 1 | - | 1 | - |
| **RV-CMi** | **0.409** | **0.022** | **0.456** | **0.002** | **0.381** | **0.011** | **0.701** | **0.0001** | **0.630** | **0.0001** | **0.708** | **0.0001** |
| **RV-GLS** | 0.24 | 0.193 | **0.488** | **0.001** | 0.131 | 0.398 | 0.219 | 0.237 | **0.497** | **0.001** | **0.348** | **0.021** |
| **RV-FWS** | 0.225 | 0.224 | 0.273 | 0.073 | 0.097 | 0.53 | 0.219 | 0.238 | **0.377** | **0.012** | 0.252 | 0.1 |
| **RV-SS** | -0.037 | 0.845 | **0.327** | **0.03** | -0.079 | 0.61 | 0.240 | 0.193 | **0.454** | **0.002** | 0.095 | 0.538 |

**B) Correlations between RV parameters and LV and RV function**

|  | LV-EF | | | | | | RV-EF | | | | | |
|---|---|---|---|---|---|---|---|---|---|---|---|---|
|  | DCM | | LVNC-R | | LVNC-N | | DCM | | LVNC-R | | LVNC-N | |
|  | r | p | r | p | r | p | r | p | r | p | r | p |
| **RV-EDVi** | -0.099 | 0.595 | **-0.314** | **0.038** | -0.200 | 0.193 | **-0.370** | **0.041** | **-0.602** | **0.0001** | **-0.369** | **0.014** |
| **RV-ESVi** | -0.212 | 0.253 | **-0.303** | **0.046** | -0.277 | 0.069 | **-0.811** | **0.0001** | **-0.868** | **0.0001** | **-0.627** | **0.0001** |
| **RV-EF** | **0.471** | **0.007** | 0.288 | 0.058 | **0.321** | **0.034** | 1 | - | 1 | - | 1 | - |
| **RV-TMi** | -0.320 | 0.079 | -0.197 | 0.246 | **-0.374** | **0.012** | **-0.626** | **0.0001** | **-0.695** | **0.0001** | **-0.542** | **0.0001** |
| **RV-GLS** | **-0.586** | **0.001** | **-0.500** | **0.001** | -0.227 | 0.138 | -0.137 | 0.463 | **-0.590** | **0.0001** | **-0.468** | **0.002** |
| **RV-FWS** | **-0.551** | **0.001** | **-0.316** | **0.037** | -0.161 | 0.298 | -0.236 | 0.201 | **-0.435** | **0.003** | -0.210 | 0.171 |
| **RV-SS** | -0.083 | 0.658 | -0.261 | 0.087 | 0.058 | 0.707 | -0.171 | 0.358 | **-0.460** | **0.002** | -0.271 | 0.075 |

**C) Correlations between RV strain values and RV volumes**

|  | RV-EDVi | | | | | | RV-ESVi | | | | | |
|---|---|---|---|---|---|---|---|---|---|---|---|---|
|  | DCM | | LVNC-R | | LVNC-N | | DCM | | LVNC-R | | LVNC-N | |
|  | r | p | r | p | r | p | r | p | r | p | r | p |
| **RV-GLS** | **0.379** | **0.035** | **0.406** | **0.006** | **0.299** | **0.049** | 0.189 | 0.308 | **0.529** | **0.0001** | 0.380 | 0.011 |
| **RV-FWS** | 0.189 | 0.308 | **0.334** | **0.027** | 0.145 | 0.348 | 0.201 | 0.279 | **0.344** | **0.027** | 0.145 | 0.348 |
| **RV-SS** | 0.286 | 0.118 | **0.315** | **0.037** | 0.022 | 0.887 | 0.256 | 0.165 | **0.451** | **0.002** | 0.152 | 0.326 |

DCM: dilated cardiomyopathy, LVNC-R: left ventricular noncompaction with reduced LV function, LVNC-N: left ventricular noncompaction with good LV function, RV-EDVi: right ventricular end-diastolic volume index, RV-ESVi: right ventricular end-systolic volume index, RV-SVi: right ventricular stroke volume index, RV-EF: right ventricular ejection fraction, RV-TMi: right ventricular end-diastolic trabecular and papillary muscle mass index, RV-CMi: right ventricular end-diastolic compact myocardial mass index, RV-GLS: right ventricular global longitudinal strain, RV-FWS: right ventricular free-wall strain, RV-SS: right ventricular septal strain, LV-TMi: left ventricular end-diastolic trabecular and papillary muscle mass index, LV-EF: left ventricular ejection fraction, r: correlation coefficient

The bold values indicate statistical significance (p<0.05).

## RV volumetric parameters

In contrast with the RV volumetric parameters of the good EF group, those of the LVNC-R and DCM groups were decreased but still in the normal range in our study, which might be due to LV enlargement that compresses the RV through mechanical interactions [18]. As potential RV involvement does not necessarily or rapidly develop in LV-affected diseases, RV function is preserved and comparable in the groups. Later, if the RV cannot compensate, the volumes might become larger, and the RV-EF might decrease [19–22]. The observed strong negative correlations between RV volumes and RV-EF also support this hypothesis.

## Ventricular trabeculation

Although there were no differences in the RV-TMi among the groups, this parameter differed significantly between the NT and HT subgroups in each patient population. The correlations of the RV-TMi with RV volumes and RV-EF corroborate the significantly higher volumes and lower RV-EF values observed in the HT subgroups with decreased LV function (LVNC-R-HT and DCM-HT) [3, 21, 23]. These volumetric and functional differences between the HT and NT subgroups have also been described in a larger noncompacted population with good LV-EF [4]. Although these significant differences in the LVNC-N group were not observed in our investigation, current correlations support the abovementioned study. These findings suggest that in the case of RV hypertrabeculation, RV function might deteriorate independently of LV function. Even with the abovementioned connections of RV function and volume and hypertrabeculation, it is still unknown whether increased volume causes higher trabeculation or vice versa, as this mechanism could be observed in all three HT subgroups. Notably, more RV-HT cases were observed in the LVNC-R group than in the DCM or LVNC-N groups, and the DCM-HT subgroup had more patients with decreased RV function than the other subgroups. A follow-up study on a larger noncompaction population would be required to verify these results, as we did not find relevant data in the literature regarding this issue. However, the prognostic role of RV function in DCM may arise in a study by Gulati et al. [24].

It is worth mentioning that RV and LV trabeculation were significantly correlated only in the LVNC-N group [4, 25]. Stacey et al. also observed a correlation between the RV apical trabecular thickness and the LV noncompacted-to-compacted ratio even in LVNC patients with reduced EF [21]. The lack of correlation in the reduced-LV-function groups might be because of the smaller sample size of our study.

As LV-TMi and LV-EF correlated with RV volumetric parameters, the former in both noncompacted groups and the latter only in the LVNC-R group, the pathological relevance of the morphological features of LVNC may arise in terms of RV.

## RV strains and LV parameters

Regarding our results, in the presence of good LV function, all the RV strains were independent of LV trabeculation and LV function. However, once LV function had decreased, both RV-GLS and RV-FWS decreased, independent of the etiology. This is supported by the literature, as more decreased RV strain values were described in DCM patients with a higher risk of major cardiovascular events combined with other cardiovascular diseases [26–30].

Not only LV function but also LV trabeculation might have an impact on RV strains: notable correlations could be observed between the RV-SS values and LV-TMi in noncompaction patients with reduced LV-EF. This could be due to ventricular interdependency, as LV contraction through the interventricular septum contributes to RV pressure by 20–40% [31].

### RV strains and RV parameters

We must highlight that relevant correlations were observed between all measured RV strain parameters and RV trabeculation only in the LVNC-R group. Moreover, significantly lower RV strains could be observed in the HT subgroup than in the NT subgroup solely in the LVNC-R population. This is in line with a larger LVNC study with good LV-EF, which also showed these correlations and differences [4]. Interestingly, the abovementioned correspondences were not observable in the DCM group. It must be emphasized that RV involvement in LVNC has a worse prognosis based on literature data [32, 33]. Regarding our results, RV trabeculation might have an impact on the deterioration of RV strains. Moreover, this suggests that the HT subgroup of the LVNC-R population is the most affected group, presumably with a worse prognosis; thus the follow-up of these patients can be recommended.

## Conclusion

To summarize, in this investigation, the characteristics of the RV and the connections between the RV and LV parameters were described, focusing on noncompaction patients with reduced LV function.

According to our results, the significant differences between patients with a normal and hypertrabeculated RV, the RV strain deterioration due to RV hypertrabeculation, and the relevant correlations in both LVNC groups could presume the involvement of the RV in noncompaction.

Our study suggests that similar to that of the LV, the progression of RV function deterioration might be assumed in noncompaction, although further studies with larger cohorts and genetic verification are required for an accurate assessment.

## Supporting information

**S1 Table. Clinical features of the studied patient groups.** DCM: dilated cardiomyopathy, LVNC-R: left ventricular noncompaction with reduced LV function, LVNC-N: left ventricular noncompaction with good LV function, ECG: electrocardiography. The bold values indicate statistical significances (p<0.05).
(DOCX)

**S2 Table. Flowchart of the enrollment process.** Exclusion criteria: *other technical reasons*: implanted devices, arrhythmic or respiratory artifacts. *comorbidities*: ischemic, valvular, or congenital heart diseases, coexisting cardiomyopathies; hypertension, diabetes mellitus, endocrine disorders, chronic kidney or systemic diseases. *intense sports activity*: >6 hours/week. LVNC: left ventricular noncompaction, DCM: dilated cardiomyopathy, LV: left ventricle, LVNC-R: left ventricular noncompaction with reduced LV function, LVNC-N: left ventricular noncompaction with good LV function.
(DOCX)

**S3 Table. Interobserver variability: The intraclass correlation coefficient was interpreted as less than 0.4, between 0.4 and 0.75, and greater than 0.75 indicated poor, fair to good and excellent interobserver agreement, respectively.** ICC: intraclass correlation coeffitients, RV-EDVi: right ventricular end-diastolic volume index, RV-ESVi: right ventricular end-systolic volume index, RV-SVi: right ventricular stroke volume index, RV-EF: right ventricular ejection fraction, RV-TotalMassi: right ventricular end-diastolic total myocardial mass index, RV-TMi: right ventricular end-diastolic trabecular and papillary muscle mass index, RV-GLS: right ventricular global longitudinal strain, RV-FWS: right ventricular free-wall strain, RV-SS:

right ventricular septal strain.
(DOCX)

**S4 Table. Comparison of the subgroups with normal RV trabeculation (NT) and with RV hypertrabeculation (HT) within the groups.** DCM-NT: dilated cardiomyopathy with normal right ventricular trabeculation, DCM-HT: dilated cardiomyopathy with right ventricular hypertrabeculation, LVNC-R-NT: left ventricular noncompaction with reduced left ventricular function and normal right ventricular trabeculation, LVNC-R-HT: left ventricular noncompaction with reduced left ventricular function and right ventricular hypertrabeculation, LVNC-N-NT: left ventricular noncompaction with good left ventricular function and normal right ventricular trabeculation, LVNC-N-HT: left ventricular noncompaction with good left ventricular function and right ventricular hypertrabeculation. RV-EDVi: right ventricular end-diastolic volume index, RV-ESVi: right ventricular end-systolic volume index, RV-SVi: right ventricular stroke volume index, RV-EF: right ventricular ejection fraction, RV-TMi: right ventricular end-diastolic trabecular and papillary muscle mass index, RV-CMi: right ventricular end-diastolic compact myocardial mass index, RV-GLS: right ventricular global longitudinal strain, RV-FWS: right ventricular free-wall strain, RV-SS: right ventricular septal strain. The bold values indicate statistical significances (p<0.05).
(DOCX)

**S1 File. Dataset.**
(XLSX)

## Acknowledgments

We thank the technicians who helped perform the cardiac magnetic resonance imaging examinations.

## Author Contributions

**Conceptualization:** Anna Réka Kiss, Hajnalka Vágó, Andrea Szűcs.

**Data curation:** Zsófia Gregor, Anna Réka Kiss, Kinga Grebur.

**Formal analysis:** Zsófia Gregor.

**Investigation:** Zsófia Gregor, Andrea Szűcs.

**Methodology:** Zsófia Gregor, Andrea Szűcs.

**Project administration:** Andrea Szűcs.

**Resources:** Andrea Szűcs.

**Supervision:** Béla Merkely, Hajnalka Vágó, Andrea Szűcs.

**Visualization:** Zsófia Gregor.

**Writing – original draft:** Zsófia Gregor.

**Writing – review & editing:** Zsófia Dohy, Attila Kovács, Hajnalka Vágó, Andrea Szűcs.

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
