## [Decision Letter · Decision Letter 0]

21 Mar 2023

PONE-D-23-04778Characteristics of the right ventricle in left ventricular noncompaction with reduced ejection fractionPLOS ONE

Dear Dr. Szűcs,

Thank you for submitting your manuscript to PLOS ONE. After careful consideration, we feel that it has merit but does not fully meet PLOS ONE’s publication criteria as it currently stands. Therefore, we invite you to submit a revised version of the manuscript that addresses the points raised during the review process.

We look forward to receiving your revised manuscript.

Kind regards,

Antoine Fakhry AbdelMassih

Academic Editor

PLOS ONE

Journal Requirements:

Reviewers' comments:

Reviewer's Responses to Questions

**Comments to the Author**

1. Is the manuscript technically sound, and do the data support the conclusions?

Reviewer #1: No

2. Has the statistical analysis been performed appropriately and rigorously? 

Reviewer #1: Yes

3. Have the authors made all data underlying the findings in their manuscript fully available?

Reviewer #1: No

4. Is the manuscript presented in an intelligible fashion and written in standard English?

Reviewer #1: Yes

5. Review Comments to the Author

Reviewer #1: The Authors present the study entitled "Characteristics of the right ventricle in left ventricular noncompaction with reduced ejection fraction". The topic could be of great clinical importance. LVNC is a disease that could worsen the clinical prognosis; therefore, describing the new indices that can help prevent clinical status deterioration would be of great interest. The Reviewer appreciates the Authors' effort and notices the promising value of the study; however, at that moment, some crucial comments need to be addressed.

Major comments:

1. The study's title is not confirmed in the text and should be changed accordingly (the Authors compared different groups of patients but not only concentrated on LVNC-R).

2. The Authors should find and precisely describe the possible clinical value of the presented results. The main differences were presented between the groups with reduced and normal LVEF; however, there were no evident differences between DCM and LVNC-R groups (that would be clinically important when RV hypertrabeculation could help to divide between that two groups).

3. Control group of persons is necessary for LVNC-N patients.

4. The presentation of the Methods section needs to be corrected (results should be excluded from that part). The Authors include some results in the Methods section- that should be changed (the results should not be included in the methods section).

5. From which patients the authors enrolled their groups retrospectively? From some Magnetic Resonance material? The flowchart with the enrollment process should be added: the Authors should write from which group of patients the enrolled groups were taken; the Authors should write precisely how many patients and why were further excluded from the analysis.

6. What is the real novelty of the presented study (in comparison to the study by Kiss AR et al. Impact of Right Ventricular Trabeculation on Right Ventricular Function in Patients With Left Ventricular Noncompaction Phenotype. Front Cardiovasc Med. 2022 Apr 12;9:843952. doi: 10.3389/fcvm.2022.843952)??? The Authors should underline that regarding their results.

Minor comments:

1. The inclusion criteria should be mentioned precisely (the authors included that information in different parts of the methods section).

1. The Authors mentioned that "All participants provided written informed consent" – how the authors obtained written informed consent in the retrospective study? That should be clarified in the methods section.

3. Information from lines 82-83 ("Patients with poor image quality or whose scans were performed after contrast agent administration or contained arrhythmic or respiratory artefacts were also excluded") should be mentioned in the magnetic resonance description.

I did not find some important citations in the manuscript:

Stämpfli SF, Donati TG, Hellermann J, Anwer S, Erhart L, Gruner C, Kaufmann BA, Gencer B, Haager PK, Müller H, Tanner FC. Right ventricle and outcome in left ventricular noncompaction cardiomyopathy. J Cardiol. 2020 Jan;75(1):20-26. doi: 10.1016/j.jjcc.2019.09.003.

6. PLOS authors have the option to publish the peer review history of their article (what does this mean?). If published, this will include your full peer review and any attached files.

Reviewer #1: No

---

## [Author Response · Author response to Decision Letter 0]

3 May 2023

Response to Reviewer

Major comments:

1. The study's title is not confirmed in the text and should be changed accordingly (the Authors compared different groups of patients but not only concentrated on LVNC-R). 

As the Reviewer requested, we have considered changing the study title to a more accurate form. We hope that the modified title reflects the manuscript’s content more precisely. Please see the revised title in the Manuscript file.

2. The Authors should find and precisely describe the possible clinical value of the presented results. The main differences were presented between the groups with reduced and normal LVEF; however, there were no evident differences between DCM and LVNC-R groups (that would be clinically important when RV hypertrabeculation could help to divide between that two groups).

The Reviewer is correct that a definitive distinction could be conducted between DCM and LVNC-R patients, in which the LV trabecular mass can be helpful, according to our results. Despite the reduced LV-EF, we need to emphasize that the average RV-EF was in the normal range in both groups in our study. Although the RV strain values of the LVNC-R and DCM groups were significantly worse than those of the LVNC-N population, significant differences between the hypertrabeculated and normally trabeculated RV subgroups could be observed only in LVNC-R. This suggests that this hypertrabeculated subgroup is more vulnerable, which might indicate an even worse prognosis.

Our results are strengthened by some studies, and the prognostic role of RV involvement in noncompaction might be predictable not only in addition to good LV function but also in the case of decreased LV-EF [1-3]. Nonetheless, the abovementioned results and the literature raise attention to the potential reduction of RV function and the importance of the follow-up of these patients.

We have emphasized the prognostic role of RV involvement in noncompaction in the introduction section (lines 60-63); moreover, the discussion section was also modified according to the abovementioned hypothesis (lines 340-341). We hope it meets the Reviewer’s expectations.

3. Control group of persons is necessary for LVNC-N patients.

We would like to thank the Reviewer for the constructive comment. As we would like to focus on noncompaction patients with reduced left ventricular function, in addition to this LVNC-R population, we would have fitted two other groups for a more comprehensive analysis: one for the noncompaction phenotype (LVNC-N group) and one for reduced LV function (DCM group).

During the study design process, the opportunity for a control group arose; however, it would be very similar to our previous studies, where the comparison of healthy controls and a large LVNC cohort with good LV function was thoroughly discussed [3, 4]. In these studies, significant differences were found between the groups in almost all LV and RV parameters, and these results were cited in our recent manuscript to give a more comprehensive picture of the role of noncompaction even with good LV function.

Thus, if we had enrolled a fourth group for controls, the manuscript could have been criticized as having redundant content, and the focus of the LVNC-R group might have been lost.

For all these reasons, with the greatest respect, we would like to refrain from involving a control group in this investigation. To meet your requirements, we emphasized the results and the comparison of the abovementioned studies even more in the discussion section (lines 282-284).

4. The presentation of the Methods section needs to be corrected (results should be excluded from that part). The Authors include some results in the Methods section- that should be changed (the results should not be included in the methods section).

The Methods section has been modified as the Reviewer needed: the Jacquier criteria were corrected for a better understanding (lines 75-76); in Table 1, the baseline characteristics were separated from LV parameters (lines 91 and 183); moreover, RV reference ranges were removed, as they were misunderstood, and only the citation was referenced (lines 98-101). Please find these changes in the modified manuscript.

5. From which patients the authors enrolled their groups retrospectively? From some Magnetic Resonance material? The flowchart with the enrollment process should be added: the Authors should write from which group of patients the enrolled groups were taken; the Authors should write precisely how many patients and why were further excluded from the analysis.

At the Heart and Vascular Center of Semmelweis University, we have an extensive register of cardiac MRI examinations conducted since 2009. We enrolled all the studied groups retrospectively from that pool. From that pool, our research group also created a highly detailed register of noncompaction patients, which helped in the most careful patient selection regarding the exclusion criteria.

The exclusion process was complex, as several criteria were met at the same time. First, all scans taken after contrast agent administration were excluded, as based on our previous investigation, these would have distorted our results [5]. Regarding the remaining LVNC population, we excluded patients with any comorbidity; then, from the rest, we selected patients with reduced LV function, to which we matched participants with good LV-EF by sex and age individually.

Regarding the DCM patients, after excluding the images taken after contrast agent administration, we selected all nonischemic, comorbidity-free DCM patients.

As the Reviewer recommended, we added a flowchart in the Supplementary material (Table S2) with the enrollment process. The exclusion criteria are also seen on page 4, lines 82-88. We sincerely hope that it will meet your requirements.

6. What is the real novelty of the presented study (in comparison to the study by Kiss AR et al. Impact of Right Ventricular Trabeculation on Right Ventricular Function in Patients With Left Ventricular Noncompaction Phenotype. Front Cardiovasc Med. 2022 Apr 12;9:843952. doi: 10.3389/fcvm.2022.843952)??? The Authors should underline that regarding their results.

The abovementioned study is one of our research team’s investigations. The main difference between the two studies is the study population. While the previous study focused on the characteristics of the RV in a highly selected LVNC population with good EF, we described the characteristics of the RV in patients with reduced LV function in the current study. We investigated how LV function deterioration affects RV volumetric, functional, and myocardial mass parameters, which were not previously detailed in the literature. The novelty of our study was that RV function was preserved in addition to reduced LV function; however, the hypertrabeculated RV had significantly worse strain values than the normally trabeculated RV. According to the references mentioned in the second question above, a worse long-term prognosis might be predictable in these patients. This hypothesis was highlighted more in the discussion section.

Minor comments:

1. The inclusion criteria should be mentioned precisely (the authors included that information in different parts of the methods section).

We would like to thank the Reviewer for reminding us of this inaccuracy. All the inclusion and exclusion criteria are mentioned in the Study population part of the Methods section; however, for the Reviewer’s recommendation, we clarified both of them and completed the Manuscript with a flowchart (lines 73 and 82 and Supplementary Material Table S2).

1. The Authors mentioned that "All participants provided written informed consent" – how the authors obtained written informed consent in the retrospective study? That should be clarified in the methods section.

The Heart and Vascular Center of Semmelweis University is an educational institution where all patients must read, fill out and sign a formal document before undergoing any examination. In this form, the patients approve that their examination’s details could be used for scientific purposes. That is the reason why we have written informed consent for a retrospective investigation. Thank you for the critical comment; we have clarified this issue in the Methods section (lines 104-105).

3. Information from lines 82-83 ("Patients with poor image quality or whose scans were performed after contrast agent administration or contained arrhythmic or respiratory artefacts were also excluded") should be mentioned in the magnetic resonance description.

In our investigation, the poor image quality, as scans were performed after contrast agent administration, or the presence of artifacts, e.g., arrhythmias and implanted devices, were limiting factors and resulted in false parameters during the postprocessing evaluation [5]. In other words, poor image quality would provide unusable data, and for this reason, these patients should be excluded from the investigation. As these circumstances do not influence the scanning procedure but determine the analyses of the population, we would like to mention these issues in the study population part, where the enrollment/exclusion processes were detailed, if it meets your requirements. Additionally, the ‘Image acquisition and analysis’ section was also completed with this information (lines 130-131).

I did not find some important citations in the manuscript:

Stämpfli SF, Donati TG, Hellermann J, Anwer S, Erhart L, Gruner C, Kaufmann BA, Gencer B, Haager PK, Müller H, Tanner FC. Right ventricle and outcome in left ventricular noncompaction cardiomyopathy. J Cardiol. 2020 Jan;75(1):20-26. doi: 10.1016/j.jjcc.2019.09.003.

We would like to thank you for bringing this article to our attention. As the Reviewer suggests, it highlights the prognostic role of distinct RV parameters in LVNC, emphasizing the importance of follow-up of these patients. As discussed above in the second question, we have completed the discussion section and cited this reference (line 338).

References

1. Stampfli, S.F., et al., Right ventricle and outcome in left ventricular non-compaction cardiomyopathy. J Cardiol, 2020. 75(1): p. 20-26.

2. Wang, W., et al., Influence of Right Ventricular Dysfunction on Outcomes of Left Ventricular Non-compaction Cardiomyopathy. Front Cardiovasc Med, 2022. 9: p. 816404.

3. Kiss, A.R., et al., Impact of Right Ventricular Trabeculation on Right Ventricular Function in Patients With Left Ventricular Non-compaction Phenotype. Frontiers in Cardiovascular Medicine, 2022. 9.

4. Kiss, A.R., et al., Left ventricular characteristics of noncompaction phenotype patients with good ejection fraction measured with cardiac magnetic resonance. Anatol J Cardiol, 2021. 25(8): p. 565-571.

5. Szucs, A., et al., The effect of contrast agents on left ventricular parameters calculated by a threshold-based software module: does it truly matter? Int J Cardiovasc Imaging, 2019. 35(9): p. 1683-1689.

---

## [Decision Letter · Decision Letter 1]

21 Aug 2023

Characteristics of the right ventricle in left ventricular noncompaction with reduced ejection fraction in the light of dilated cardiomyopathy

PONE-D-23-04778R1

Dear Dr. Szűcs,

We’re pleased to inform you that your manuscript has been judged scientifically suitable for publication and will be formally accepted for publication once it meets all outstanding technical requirements.

Kind regards,

Antoine Fakhry AbdelMassih

Academic Editor

PLOS ONE

Additional Editor Comments (optional):

Please during proofing, move Table 1 to results section 

Reviewers' comments:

Reviewer's Responses to Questions

**Comments to the Author**

1. If the authors have adequately addressed your comments raised in a previous round of review and you feel that this manuscript is now acceptable for publication, you may indicate that here to bypass the “Comments to the Author” section, enter your conflict of interest statement in the “Confidential to Editor” section, and submit your "Accept" recommendation.

Reviewer #1: (No Response)

2. Is the manuscript technically sound, and do the data support the conclusions?

Reviewer #1: Yes

3. Has the statistical analysis been performed appropriately and rigorously? 

Reviewer #1: Yes

4. Have the authors made all data underlying the findings in their manuscript fully available?

Reviewer #1: Yes

5. Is the manuscript presented in an intelligible fashion and written in standard English?

Reviewer #1: Yes

6. Review Comments to the Author

Reviewer #1: The Authors corrected their text nicely according to my previous comments. It would be helpful whether the Authors write in red color their new elements of the text additionally to information regarding lines.

For that moment I have only one minor comment:

The Authors should exclude ALL the results from the Methods section. (Table 1 present the results).

7. PLOS authors have the option to publish the peer review history of their article (what does this mean?). If published, this will include your full peer review and any attached files.

Reviewer #1: No

---

## [Editor Report · Acceptance letter]

15 Sep 2023

PONE-D-23-04778R1 

Characteristics of the right ventricle in left ventricular noncompaction with reduced ejection fraction in the light of dilated cardiomyopathy 

Dear Dr. Szűcs:

I'm pleased to inform you that your manuscript has been deemed suitable for publication in PLOS ONE. Congratulations! Your manuscript is now with our production department. 

Kind regards, 

on behalf of

Prof Antoine Fakhry AbdelMassih 

Academic Editor

PLOS ONE